# Electrostatics in Computational Biophysics and Its Implications for Disease Effects

**DOI:** 10.3390/ijms231810347

**Published:** 2022-09-07

**Authors:** Shengjie Sun, Pitambar Poudel, Emil Alexov, Lin Li

**Affiliations:** 1Computational Science Program, The University of Texas at El Paso, 500 W University Avenue, El Paso, TX 79968, USA; 2Department of Physics and Astronomy, College of Science, Clemson University, Clemson, SC 29634, USA; 3Department of Physics, The University of Texas at El Paso, 500 W University Avenue, El Paso, TX 79968, USA

**Keywords:** electrostatics, computational biophysics, disease mechanism, mutations, pH-dependence, protein–protein interactions, protein stability

## Abstract

This review outlines the role of electrostatics in computational molecular biophysics and its implication in altering wild-type characteristics of biological macromolecules, and thus the contribution of electrostatics to disease mechanisms. The work is not intended to review existing computational approaches or to propose further developments. Instead, it summarizes the outcomes of relevant studies and provides a generalized classification of major mechanisms that involve electrostatic effects in both wild-type and mutant biological macromolecules. It emphasizes the complex role of electrostatics in molecular biophysics, such that the long range of electrostatic interactions causes them to dominate all other forces at distances larger than several Angstroms, while at the same time, the alteration of short-range wild-type electrostatic pairwise interactions can have pronounced effects as well. Because of this dual nature of electrostatic interactions, being dominant at long-range and being very specific at short-range, their implications for wild-type structure and function are quite pronounced. Therefore, any disruption of the complex electrostatic network of interactions may abolish wild-type functionality and could be the dominant factor contributing to pathogenicity. However, we also outline that due to the plasticity of biological macromolecules, the effect of amino acid mutation may be reduced, and thus a charge deletion or insertion may not necessarily be deleterious.

## 1. Introduction

In this review, we consider the role of electrostatics on the structure and function of biological macromolecules and summarize the specific features that manifest electrostatic interactions. We refer to electrostatic interactions that result from charged atoms, including the atoms of water molecules. This allows us to attribute these specific features and their disruption caused by missense mutations to molecular mechanisms implicated in diseases. We focused on computational findings; however, when possible, these findings are backed up by experimental observations. The review emphasizes recent investigations to assess current interest in this field. 

The electrostatic features in biological molecules are described in this review within three functions: long-range effects [1,2]; short-range effects [3,4,5]; and pH-dependent effects [6,7,8]. The long-range electrostatic force is the main force making biomolecules travel and bind to other biomolecules over a long distance [1,9]. Due to the limitations of current computing, it is challenging to simulate long-range electrostatics in molecular dynamic simulations. Therefore, many efforts have been proposed and developed for simulations involving long-range electrostatic interactions [2,9]. Long-range effects are introduced in Section 2.1. Such algorithms of long-range electrostatics are widely used in biological problems, such as protein–protein interactions [10,11,12], protein–DNA/RNA interactions [9,13], etc. Short-range interactions, including salt bridges and hydrogen bonds, which are favorable interactions in terms of Coulombic energy, are present inside/among the macromolecules, serving as an important feature contributing to macromolecular architecture. Short-range effects frequently contribute to receptor–ligand binding [14,15] and to the specificity of the binding mode [16,17]. The pH-optimum is the particular pH at which biological activity, macromolecular stability, and binding are best optimized [18]. It originates from the pKa values of the ionizable groups, indicating the importance of assigning correct protonation states in molecular dynamics simulations [19,20,21,22,23]. Recent contributions to modeling pH-dependent phenomena are shown in Section 2.3. Constant pH MD (CpHMD) approaches are an efficient tool to explore the effects of pH on biomolecular stability [24] and binding [25].

Electrostatics plays a crucial role in biomolecular stability and binding. Thus, mutations causing deviation of electrostatic properties may influence macromolecular thermodynamics, resulting in dysfunctional biomolecules. Mutations can either delete or reverse a charge, and these changes are expected to cause serious effects on both long- and short-range interactions [26,27]. Recent works on salt-bridge disruption and hydrogen-bond disruption [28,29] are shown in Section 3. The last section is about drug discovery, and we focus on computational works that indicate the role of salt bridges and hydrogen bonds [30,31,32,33] to identify efficient drugs. This includes inhibitor screening methods for hepatitis C virus [34,35], insulin amyloid fibril [36], and antibiotics against SARS-CoV-2 [37,38]. The goal of this review is to highlight the role of electrostatics in the field of computational biophysics and related disease mechanisms. Electrostatics is related to various molecular mechanisms for wild-type functionality and disease causality [27,39]. Electrostatic features are the key component in drug design according to short- and long-range effects. Another prominent representation of the role of electrostatics is pH-dependence, which typically originates from both long- and short-range electrostatic interactions. Overall, many biological processes at the molecular level are strongly affected by electrostatics, which should be seriously considered in any atomistic-level studies.

## 2. Electrostatics of Wild-Type Biological Macromolecules

The role of electrostatics in wild-type biological macromolecules is considered on three levels: (a) long-range effects, i.e., interactions that do not involve physical contact between interacting entities; (b) short-range effects, i.e., interaction involving physical contacts, such as the formation of salt bridges or hydrogen bonds; and (c) pH-dependent phenomena, which involve a mixture of long- and short-range effects (Figure 1).

### 2.1. Long-Range Electrostatic Effects

The electrostatic force is a long-range force that dominates all other forces when there is no physical contact between the molecules. It is also a universal force in molecular biology since all atoms carry a partial charge and thus are subject to electrostatic interactions. Because of this, electrostatics are expected to play a key role in steering interacting partners toward their binding position (Figure 1a,b—electric field lines demonstrate long-range attraction between the partners). However, computational modeling of such a process, especially if the interacting partners are far apart, is quite computationally demanding. Indeed, in typical molecular dynamics (MD) simulations, one applies cut-offs of long-range interactions to speed up the calculations [1]. This is valid also for protein folding studies, particularly modeling of the unfolded state. Recent work demonstrated that the unfolded state’s structural characteristics are strongly dependent on the cut-offs and Evald sum implementation [47] (a method for computing long-range interactions).

To overcome such limitations, a hybrid method involving DelPhi [48,49,50,51] (a popular Poisson–Boltzmann equation solver) and steered MD was developed. The method, termed DelPhi–Force MD (DFMD) [10,52], was successfully applied to dock substrates to an enzyme [2] and a biomolecule onto another biomolecule (such as barstar onto barnase) [10]. It was demonstrated that such an approach allows one to take advantage of the MD protocol to sample different conformations while simultaneously adding electrostatic force-based guidance of the ligand toward the correct binding pocket/interface [2]. Moreover, it was shown that DFMD can successfully dock to barnase even if the initial positions and orientations of both are completely different from the correct ones [10].

Of particular interest are cases involving highly charged large biological objects such as microtubules. Microtubules are highly negatively charged long objects that serve as highways for various transport (cargo) proteins such as kinesin and dynein. The cargo proteins walk on microtubules, and one could expect that their walking is influenced by long-range electrostatic interactions. Indeed, it was computationally shown that there are strong electrostatic interactions between microtubules and microtubule binding domains of both kinesin [53] and dynein [54]. The same observation was made for G-actin and myosin (Figure 2c). These computational findings were experimentally confirmed in a study of a series of mutants at the dynein Microtube Binding Domain (MTBD)–microtubule-binding interface to neutral residues. It was discovered that the altered MTBDs’ binding affinity for microtubules was significantly increased. Furthermore, it was discovered that the binding and unbinding rates of MTBDs to microtubules were significantly impacted by charge screening of free ions in solution. These findings show the importance of long-range electrostatic interactions in controlling dynein–microtubule affinity [55].

The role of electrostatics in providing a guiding force for macromolecular binding was demonstrated in a series of examples using the experimental structure of the complex and calculating the electrostatic force generated by one of the partners on the other one. The examples included homodimers, calmodulin, protein–DNA/RNA complexes, and quinone in the reaction center protein [56]. Furthermore, the electrostatic interactions were found to be a crucial factor for virus assemblies (Figure 2a). Many studies have proven that the electrostatic interactions among capsomers of the virus provide attractive forces for the viral capsid assembly [57,58].

Electrostatic force not only guides the binding partners toward their association, but it also assures that their interfaces are properly oriented [52] (Figure 2b). This is demonstrated in cases in which the binding partners are deliberately oriented improperly prior to simulations, such that their interfaces are not aligned. The electrostatic force generates torque that reorients the partners to a proper orientation. This has been demonstrated in a series of examples, including barnase–barstar [10], dynein–microtubule [12], and capsomer–capsomer interactions in viral capsid assembly [58].

As mentioned above, long-range electrostatic interactions provide a guiding force that drives the partners together; however, this force is not uniform and changes as a function of the distance between the partners. It was demonstrated that electrostatic force profiles (force as a function of distance) can be grouped into four distinctive categories [11]. In some cases, electrostatics favor the binding; in others, they oppose it. The most intriguing case is termed the “soft landing” case such that at large distances electrostatic force attracts the partners, but just prior to physical contact, the electrostatics oppose the binding and assure soft binding [11] (Figure 2d). Soft landing may help two binding partners contact each other in sufficient time, so that they can adjust the configurations at the binding interfaces to avoid atom–atom clash and can form hydrogen bonds and salt bridges.

Of particular interest are cases involving intrinsically disordered proteins (IDP). Thus, computational and experimental analysis of the cell-cycle regulator p27 have demonstrated that long-range electrostatic forces acting on enriched charges of IDPs can speed up protein–protein encounter through “electrostatic steering”, and can simultaneously promote “folding-competent” encounter topologies to increase the efficiency of IDP folding upon encounter [60]. Another study of three IDP-forming complexes also outlined the importance of electrostatics. The authors conducted topology-based coarse-grained simulations and projected identical electrostatically accelerated encounter and folding mechanisms for all three complexes. These findings were consistent with earlier research on the charge distributions in known IDP complexes, which indicates that electrostatic interactions play a significant role in facilitating effective coupled binding and folding for quick specific identification [61]. Similarly, computational alanine scanning has been performed for SARS-CoV-2 Spike and Human ACE2 proteins. The residues identified with significance to binding and other proximal residues were studied further through molecular mechanics-based protein-binding free-energy-change prediction [62].

### 2.2. Short-Range Electrostatic Effects

At large distances between interacting atoms, electrostatics manifest their role via direct charge–charge interactions; in the case of short distances when there is contact between interacting partners, electrostatic contributions are more complex. They still involves direct charge–charge interactions and solvation energy. Direct charge–charge interactions are very strong at Angstrom scale distance (strictly speaking, short-range effects are the results of electrostatic interactions, orbital relaxation, and Pauli-repulsion). In the terminology of continuum electrostatics, the balance is maintained by Coulombic interactions and desolvation penalty. Thus, a salt bridge or a hydrogen bond, which are favorable interactions in terms of Coulombic energy, may not contribute to the macromolecular stability or binding due to overcompensation coming from desolvation penalty (Figure 1c,d: the salt bridge pair of ASP326–ARG37 is buried inside the protein, while for GLU205–LYS203 it is on the surface of the protein. The hydrogen bonds of THR445–THR276 and ASN397–TYR421 are inside the protein, while those of GLU160-SER155 are surface-exposed). However, such unfavorable salt bridges and hydrogen bonds may still be present in the macromolecular universe and are an important structural feature contributing to macromolecular architecture. Such short-range interactions are frequently found to contribute to receptor–ligand binding and to the specificity of the binding mode. Below we outline such short-range electrostatic interactions for specific cases found computationally and, if possible, backed up with experimental observations.

Polarization of interacting atoms is particularly important when the interactions are short-range. Thus, the impact of electronic polarization on homogeneous and heterogeneous amyloid oligomers using the Drude Oscillator model (a model considering each atom to be a point particle with a partial charge during the simulation) was recently investigated. The system of interest was made of two amyloids, amyloid-β (Aβ) and islet amyloid polypeptide (IAPP), which are involved in the pathology of Alzheimer’s disease [63] and type 2 diabetes mellitus [64]. It was found that the most stable system was homogenous A-16-22, which gained stability from salt-bridge formation and reduced polarization in hydrophobic residues [62]. Similarly, another study investigated β-strand rich oligomers with Drude force field and found that structural rearrangement occurred, causing some loss of β-strand structure in favor of random coil content for all oligomers [16]. It was outlined that low polarization in hydrophobic residues and salt-bridge formation contribute to the stability of homogenous Aβ_16−22_ [16].

Short-range interactions are a key factor that contributes to macromolecular stability, and therefore have received significant interest. The interplay is between favorable direct interactions and desolvation penalty. Thus, a recent study reported the role of two salt bridges in ubiquitin: surface-exposed salt bridge (SB1:K11-E34) and buried salt bridge (SB2:K27-D52). It was shown that if SB1 is broken, the mechanical stability of ubiquitin increases slightly, but when SB2 is broken, the stability reduces significantly [65]. Stability and activity are typically related, and thus a study on a particular G-protein coupled receptor (GPCR), the cannabinoid receptor 1 (CB1), indicated that the switching mechanism involves a salt bridge, the D263–K328 salt-bridge, that contributes to the stability of CB1 [66]. Salt-bridge involvement in the activation mechanism of calcium-dependent protein kinase-1 further elucidates the role of short-range interactions in molecular biology. Recent work focused on calcium-dependent protein kinase-1 of Plasmodium falciparum (PfCDPK1) and explored the possibility of allosteric inhibition of this kinase. It showed how the truncation of CAD in PfCDPK1 disrupts a conserved salt-bridge required for stabilizing the kinase domain in an active state, and the kinase domain adopts an inactive conformation [66]. Similarly, the role of a salt bridge, His59–Asp103, in human granzyme B (hGzmB) was studied. It was shown that Asp103–Arg216 forms a salt bridge upon activation, breaking the His59–Asp103 hydrogen bond and enlarging the active site to aid in substrate binding [67]. Furthermore, short-range interactions are implicated in protein–protein binding, and mutations may be disease-related. This was investigated to determine if SARS-CoV-2 infectivity is enhanced by naturally occurring mutations. The structural dynamics of the RBD spike protein mutation enhancing ACE2 binding were computed in silico to achieve this. Due to better interfacial stability of the RBD strand surrounding the ACE2 across salt bridge hotspots, the interfaces in the RBD region showed stronger affinity for ACE2 [14]. A related study investigated mutations (S477N–E484K) in the receptor-binding domain (RBD) of the spike protein. The binding affinity of ACE2–RBD was examined using protein docking and all-atom simulation. According to the investigation, the mutant modifies the hydrogen bonding network and binds more strongly than the wild-type [14].

Short-range interactions are essential for gating and transport, which are frequently involved in selectivity. This was demonstrated in the case of the potassium channel, where it was observed that S42 mutations in the pore helix significantly slow the shutting of this filter gate, an effect that is unrelated to the amino acid’s creation of a hydrogen bond at this location [68]. The role of salt bridges in conformational changes needed for the mitochondrial ADP/ATP carrier (AAC), which alternates between cytosol-open (c-) and matrix-open (m-) states to export ATP and import ADP, was investigated via molecular dynamics simulations. It was shown that short-range interactions are critical for the functioning of the carrier [69].

Hydrogen bonds are frequently involved in catalysis and the overall function of biological macromolecules. Thus, it was demonstrated that the wild-type hydrogen bond network is crucial for the activity of alpha/beta hydrolase domain-containing 5 (ABHD5), also known as CGI-58, which is the activator of adipose triglyceride lipase (ATGL) [39]. Indeed, computational modeling showed that mutations to E41, R116, and G328 disrupt the hydrogen bonding network with surrounding residues and inhibit membrane targeting or ATGL activation [70]. Another study focused on the T cell receptor’s interaction with the peptide major histocompatibility complex and showed that hydrogen bonds and Lennard–Jones contacts, which are physicochemical aspects of the TCR–pMHC dynamic bond strength, are correlated with the immunogenic response brought on by the particular peptide in the MHC groove [71]. Hydrogen bonds are essential in catalytic reactions involving proton transfer. This was demonstrated for the breakdown of uracil, in which the nucleoside triphosphate cyclohydrolase (UrcA) catalyzes the two-step hydrolysis of uridine triphosphate (UTP). MD simulations showed that hydrogen bond interaction helps the reaction intermediate undergo spontaneous conformation overturn in the active site of UrcA [72].

Perhaps hydrogen bonds are the most important for water molecule arrangement. Since the physiological environment of all biological macromolecules is the water phase, the properties of the water medium affect their stability and functionality. This inspired investigations on understanding short-range interactions between neighboring water molecules. Since the interactions occur at very short distances, one needs to apply quantum mechanical approaches. Thus, recent work focused on the implications of nuclear quantum fluctuations on equilibrium and dynamical properties relating to bifurcation routes in hydrogen-bond dimers of water and ammonia. It was shown that the classical over-the-hill approach is substituted with a tunneling-controlled mechanism that, from the perspective of the path integral, can be modeled as coordinated inter-basin migrations of polymer beads [73]. The application of quantum mechanical approaches was further extended to probe the development of neighboring hydrogen bonds with water molecules. It was demonstrated that there is a direct mechanism for the emergence of short-range structural fluctuations in the hydrogen bond network of liquid water, which shows that the time development of neighboring hydrogen bonds is closely connected [74,75].

Hydrogen bonds and salt bridges are important for receptor–ligand interactions, including drug-like molecules binding to their corresponding targets. Recent work explored this topic and applied MD simulations with MMPBSA approaches to study the stability of histone deacetylase inhibitors’ effects on the stability of histone deacetylase-like proteins (HDLP). The result showed that the stability of HDLP–CBHA (m-Carboxycinnamic acid bis-hydroxide (CBHA)) is higher than that of the free HDLP enzyme. The higher stability was contributed by the increased number of hydrogen bonds [15]. Similarly, computational modeling of a prospective drug, dorzagliatin, showed it interacting with human glucokinase activator (GKA). The results showed that dorzagliatin can create the characteristic hydrogen bonds of GKA with Arg63, making a tightly binding hydrogen bond network around dorzagliatin [76]. Even in the case of DNA, a computational study indicated that the hydrogen bonds, being the main interaction constraining the variability of the linkers, are enhanced slightly with DNA twist number [77]. Another study carried out MD simulations to study riboswitches and found that the binding domain in riboswitch is stable for molecule recognition and binding, and the switching base pairings are co-evolved in the translation state [78]. Electrostatic interactions were shown to be a crucial factor for reversing dysfunctional p53 and thus avoiding tumor progression. The N-terminal transactivation domain (TAD) of p53 can regulate cell apoptosis by interacting with the transcriptional adaptor zinc-binding 2 (Taz2) domain of p300. It was demonstrated that electrostatic interactions govern the affinity of the p300 Taz2–p53 TAD2 complex [17].

### 2.3. pKa Calculations and pH-Dependent Phenomena

Almost all biological processes are pH-dependent, which indicates the crucial role of hydrogen ion concentration in living systems [79,80]. This is manifested as pH-dependence of catalysis, stability, binding affinity, and conformational flexibility [80,81]. Typically, there is a particular pH at which biological activity, macromolecular stability, and binding are best optimized, termed pH-optimum [18]. The pH-dependence originates from the pKa values of the ionizable groups, both ionizable amino acids and nucleic acids, and thus indicates the importance of assigning correct protonation states before any modeling or introducing of pH in molecular dynamics simulations (constant-pH MD [19,20,21,22,23]). Below, we outline recent contributions to modeling pH-dependent phenomena. It is well-understood that pH and conformational changes are coupled, and in recent times, many researchers have applied constant-pH MD (CpHMD) approaches to model the effect of pH on protein stability [24], binding [25], dynamics [82], and reactivity [83]. Additional complexity, if one investigates membrane proteins, comes from the presence of the lipid bilayer [84]. Thus, the pKa values of ionizable groups may undergo pKa shift as proteins insert into the membrane [85]. In parallel, a significant amount of work has been completed using either predefined conformational space [86] or mimicking the conformational stability via Gaussian-based smooth dielectric function [87]. Furthermore, some methods use coarse-grained lattice-based models of proteins and train the model on existing experimental data [88] and the treecode-accelerated boundary integral solver [89].

In terms of long- and short-range electrostatic interactions discussed above, the pH-dependent phenomena originate from a mixture of both. Thus, a recent computational study of a particular protein (Apolipophorin-III) that associates with a lipid disk demonstrated that the association is strongly pH-dependent, but there are no direct interactions between the titratable group of Apolipophorin-III and charged lipid head groups [90]. In contrast, another study on cyclic dipeptides showed that the pKa of side chains of lysine increases for cyclic dipeptides compared to the linear ones, which was attributed to short-range interactions [91]. The interplay between short- and long-range electrostatic interactions was demonstrated in a study of virus assembly. The authors outlined that both the stability and the binding affinity of the E protein are pH-independent in pH range of 6 to 10, and that this is a result of a network of interactions [92]. The short- and long-range interactions are specifically important in treating the water phase, either explicitly or implicitly, particularly when one deals with water channels or wires inside biological macromolecules, where the water flexibility is highly reduced as compared with bulk water. Thus, water molecules in the Gramicidin A (gA) channel were investigated to probe the effect of different approaches and computational techniques. It was concluded that MCCE and Drude analysis led to a small net dipole moment as the water molecules changed orientation within the channel [93].

The pH-dependence is also pronounced in allosteric regulation. A recent work reported combined experimental with computation investigations and showed that allostery in pH-switching proteins is guided by coupling throughout the protein, featuring a large network of hydrophobic interactions that work in concert with key electrostatic interactions [94].

## 3. Electrostatics and Disease Mechanisms

It has been outlined that electrostatics play a crucial role in macromolecular stability and binding. Therefore, any deviation from wild-type electrostatic properties would have profound effects on macromolecular thermodynamics and may thus result in a dysfunctional biological macromolecule. Furthermore, if the biological macromolecule is important for the wild-type functioning of the cell, its dysfunction could cause disease. Surely, as demonstrated in the literature, there is a linkage between the effect of missense mutations on protein stability and/or interactions and the propensity of the mutation to be photogenic [95,96]. Mutations that either delete a charge or reverse a charge are expected to cause a dramatic effect on both long- and short-range interactions and thus to have a high propensity to be disease-causing [26,27]. Below, we outline recent papers that model the effect of disease-causing mutations on the atomic scale, including disruption of salt bridges and hydrogen bonds, and in silico design of drugs that could mitigate the disease-causing effect [28,29].

### 3.1. Salt-Bridge Disruption

In the previous section, we emphasized the important role that salt bridges play in stability and interactions of biological macromolecules. However, it should be clarified that the expectation is that disruption of buried salt bridges should be more deleterious than that of surface-exposed ones (unless they participate in binding a partner). Even more, due to the plasticity of biological macromolecules, even a mutation within a buried salt bridge may not cause a large change in the stability because the macromolecule can rearrange and accommodate the change [26].

Since electrostatic interactions are long-range interactions, the deleterious effects may involve charged amino acids situated far apart from each other. Thus, a recent paper modeled the electrostatic component of the force acting between a kinesin motor domain and tubulin. The receiver–operating characteristic method is used to show that variations in the electrostatic component of the binding force can distinguish between disease-causing and non-disease-causing mutations detected in the human kinesin motor domain. The prediction rate of 0.843 area under the ROC curve owing to a change in the amplitude of the electrostatic force alone is notable, because diseases may result from a variety of causes unrelated to kinesin–microtubule binding [97]. Disruption of intermolecular interactions was implicated in another disease, prion disease. It was proposed that pairs of amino acids from opposing subunits form four salt bridges to stabilize the zigzag interface of the two protofibrils (Figure 3a). The results provided structural evidence of the diverse prion strains and highlighted the importance of familial mutations in inducing different strains. Electrostatics play an essential role in the formation of other complexes, such as the virulence factor ESAT-6. This is related to mycobacterium tuberculosis (Mtb), which is a leading death-causing bacterial pathogen. ESAT-6 is hypothesized to form an oligomer for membrane insertion of Mtb as well as for rupturing [98].

### 3.2. Hydrogen Bond Disruption

Hydrogen bonds are essential for maintaining the 3D structure of biological macromolecules and the formation of macromolecular complexes. Disruption of wild-type hydrogen bonds by mutations may or may not be deleterious (Figure 3b). The same considerations apply as outlined for salt bridges above. Below, we outline recent works on this subject.

The binding and dissociation between myosin and actin filaments are crucial for heart protection and drug development. It has been demonstrated that the loss of hydrogen bonds significantly promotes the detachment of myosin from actin filament, causing state changes from a rigor state to a post-rigor state [5]. Polyglutamine tracts are regions of low sequence complexity frequently found in transcription factors. The length of tracts is related to transcriptional activity, and expansion beyond specific thresholds is the cause of polyglutamine disorders. The conformation of the polyQ tract of the androgen receptor is associated with spinobulbar muscular atrophy, depending on its length. This sequence folds into a helical structure, which is stabilized by unconventional hydrogen bonds between glutamine side chains and main chain carbonyl groups. Its helicity directly correlates with tract length. These unusual hydrogen bonds are bifurcated with the conventional hydrogen bonds stabilizing α-helices [102]. TDP-43 is an essential RNA-binding protein forming aggregates in nearly almost all cases of sporadic amyotrophic lateral sclerosis (ALS), frontotemporal lobar dementia (FTLD), and other neurodegenerative diseases. TDP-43 aggregates have a self-templating, amyloid-like structure. This segment adopts a beta-hairpin structure that forms an amyloid-like structure. This conformer is stabilized by a special class of hyper-cooperative hydrogen bonding [103]. Moreover, amyloid fibrils of α-synuclein are the histological hallmarks of Parkinson’s disease, dementia with Lewy bodies, and multiple-system atrophy. The H50Q mutation results in two novel polymorphs of α-synuclein: narrow and wide fibrils, formed from either one or two protofilaments, respectively. These structures reveal new structural elements, including a hydrogen-bond network and surprising new protofilament arrangements [104].

### 3.3. pH-Dependence Alteration

Biological macromolecules have evolved to function at a particular pH, and any deviation of wild-type pH-dependent properties on stability and binding may cause diseases. Such changes can be caused by mutations involving titratable groups and, less frequently, by mutations causing conformational changes and thus affecting the pKa of titratable residues. Even a mutation that involves same-polarity residues, such as Arg to His, could be disease-causing (Figure 3c). Indeed, since the intracellular pH of most cancers is constitutively higher than that of normal cells and enhances proliferation and cell survival, the substitution of Arg with His dramatically changes the activity of mutant proteins at high pHs [105,106,107].

The altered pH-dependent properties of biological macromolecules are manifested not only in cancer but also in many other diseases. Thus, a recent work computationally studied the pH-dependent stability of several melanosome membrane proteins. The authors found that disease-causing variants impact the pH-dependence of melanosome proteins [8]. Substitution of wild-type His with Gln was found to alter the function of CLIC2 protein and to cause X-linked channelopathy with cardiomegaly [108].

The COVID-19 pandemic prompted many investigations on the molecular mechanism of disease and the effect of mutations. Recent work found that severe acute respiratory syndrome coronavirus 2 (SARS-CoV-2), a causative agent of the COVID-19 pandemic, is thought to release its RNA genome at either the cell surface or within endosomes, the balance being dependent on spike protein stability and the complement of receptors, co-receptors, and proteases. The investigator performed pKa calculations on a set of structures for spike protein ectodomain and fragments from SARS-CoV-2 and other coronaviruses. It has been predicted that a particular aspartic acid contributes to a pH-dependence of the open/closed equilibrium [109]. Another work applied multi-scale computational approaches to study the electrostatic features of spike (S) proteins for SARS-CoV and SARS-CoV-2. The authors demonstrated that the complex structures of hACE2 and the S proteins of SARS-CoV/SARS-CoV-2 are stable at pH values ranging from 7.5 to 9 [6]. A plausible route in which variants can alter viral properties is the transition between receptor binding domain (RBD) up and down forms of the SARS-CoV-2 spike protein trimer. The work predicted that pH-dependence in the mild acidic range, with stabilization of the locked form as pH, reduces from 7.5 to 5 [110].

## 4. In Silico Drug Discovery

Here we do not attempt to outline the work in the general field of drug discovery; rather, we focus on investigations of molecular effects of disease-causing mutations or the seeking of inhibitors from natural compounds (Figure 3d). In the latter case, we emphasize works that manifest the role of salt bridges and hydrogen bonds to identify efficient inhibitors.

Once the effect of a disease-causing mutation is revealed, it is tempting to seek a small molecule, a potential drug, which can mitigate the effect; this frequently involves electrostatic alteration. This is affordable because genotypes aggregate into several phenotypes only. Thus, a particular drug is supposed to be efficient for many genotypes within the same gene. Indeed, in the case of Rett syndrome, the majority of the most frequent disease-causing mutations have been experimentally and computationally shown to lower the affinity of MeCP2 to the cognate DNA, while not affecting protein stability [111,112]. Similarly, many genotypes, meaning many mutations in the spermine synthase gene that cause Snyder–Robinson syndrome, were found to affect spermine synthase homo-dimerization, and thus the electrostatic funnel that guides the substrate to the active site [30,31,32,33]. This resulted in investigations that proposed drugs capable of mitigating disease-causing effects [28,29,113]. The most common approach to developing drugs is to design inhibitors. Such a design requires considering long- and short-range electrostatic effects. A recent study carried out residue study to investigate tyrosine kinase enzyme inhibitors. The result showed that a pair of aspartic residues, providing negative potential, plays an important role in providing attractive interactions in the binding site of the enzyme [114].

Hepatitis C virus (HCV) affects millions of people worldwide. Existing drugs have different efficiencies against different genotypes of HCV. This prompted a computational investigation to select the most efficient inhibitor against the most prevalent genotype in South Asia; it was shown that short-range hydrogen bonds contribute the most to the binding energy [35]. Molecular docking was utilized in another study to identify inhibitors of insulin amyloid fibril formation. It was elucidated that hydrogen bonds play a crucial role in the interactions between the ligands and insulin [36]. Molecular dynamics and MMPBSA methods were also used to probe the effect of inhibitors. This was done in the case of histone deacetylase-like proteins, and showed that affinity increases with the increase in hydrogen bonds [15]. Similarly, inhibitors against cyclin G-associated kinase involved in hepatitis C virus entry into host cells were studied, and short-range interactions were analyzed [34]. The importance of hydrogen bonds was demonstrated in a study of amoxicillin, widely known as an antibiotic, to bind to COVID-19 protein in Mpro protease [37]. Another study focused on the observation that SARS-CoV-2 entrance into the host cells occurs via interactions between the receptor binding domain (RBD) of the spike (S) protein on the virus with the angiotensin-converting enzyme 2 (ACE2) receptor. Thus, the authors took a peptide consisting of residues 19–48 of ACE2 as the wild-type peptide, along with six mutants. They showed that short-range interactions are essential for selecting the best compound [38]. Other plausible inhibitors of SARS-CoV-2 were also suggested by targeting proteins crucial for SARS-CoV-2 infection and cytokine storm [115]. The role of electrostatic interactions was demonstrated in another study regarding inhibitors against DNA methyltransferase [116]. Short-range interactions were analyzed to identify inhibitors against acetylcholinesterase, which is a key enzyme enhancing cognitive disorder and leading to Alzheimer’s disease [117]. Aminoquinolines bind to hemoglobin, and thus prevent the degradation of hemoglobin, while by binding to parasitic tissue, aminoquinolines diminish the strength of the parasite that causes malaria. The role of hydrogen bonds in these association events was investigated and showed that the number of hydrogen bonds differs depending on the target [88].

## 5. Conclusions

This review outlined recent contributions emphasizing the role of electrostatics in the field of computational molecular biophysics and disease mechanisms. It outlined that electrostatics are still a hot topic, and it is implicated in various molecular mechanisms for wild-type functionality and disease causality. Furthermore, electrostatic considerations are frequently the key component in drug design to mitigate the disease mechanism. While an attempt was made to group the contributions according to short- and long-range effects, due to the complex nature of electrostatic interactions, both cases are often present in biological systems. A prominent manifestation of the role of electrostatics in molecular biology is pH-dependence, which typically originates from a mixture of long- and short-range electrostatic interactions. Overall, electrostatics appear to be involved in many processes at the molecular level and should be considered in any study dealing with an atomistic level of details.

## Figures and Tables

**Figure 1 ijms-23-10347-f001:**
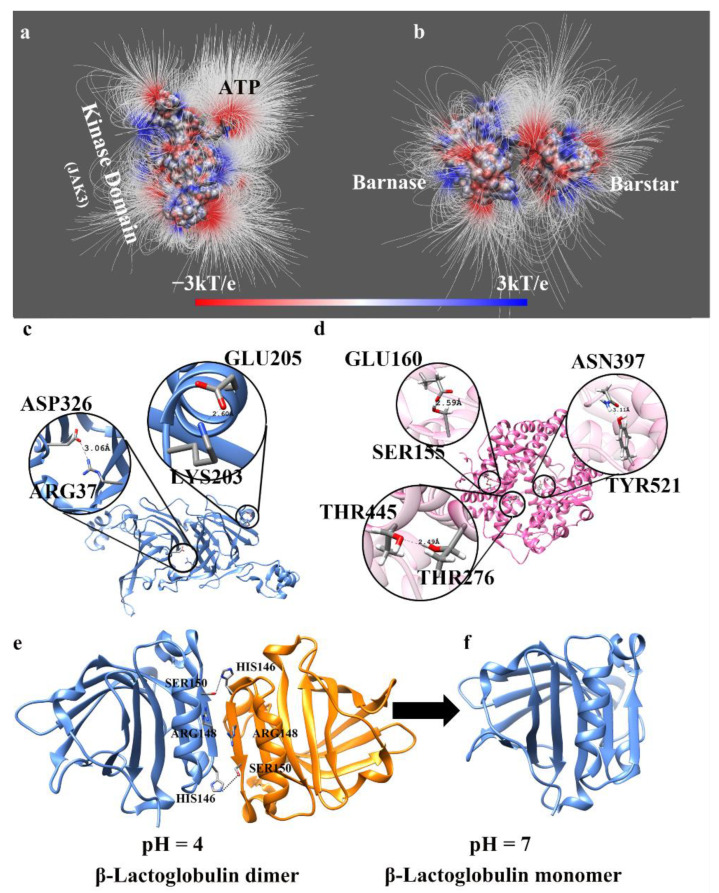
(**a**) The electric field lines between the JAK3 kinase domain and ATP. The kinase domain and ATP were separated by 15 Å to show the interactions [40] (the structure is constructed by I-TASSER [41] to model the missing loops). (**b**) The electric field lines between barnase and barstar (PDB: 1BRS [42]). The two proteins were separated by 10 Å to show the interactions. (**c**) The salt bridges in 8S α-Globulin (PDB: 2CV6 [43]). The missing loops of the 8S α-Globulin are modeled by SWISS-MODEL [44]. The salt-bridge pair of ASP326–ARG37 is buried inside the protein, while GLU205–LYS203 is on the surface of the protein. (**d**) The hydrogen bonds in Angiotensin-Converting Enzyme 2 (ACE2) (PDB: 6LZG [45]). The hydrogen bonds of THR445–THR276 and ASN397–TYR421 are inside the protein, while those of GLU160–SER155 are surface-exposed. (**e**) The β-Lactoglobulin dimer formed at pH 4. (**f**) The β-Lactoglobulin monomer formed at pH 7 (PDB:6FXB [46]).

**Figure 2 ijms-23-10347-f002:**
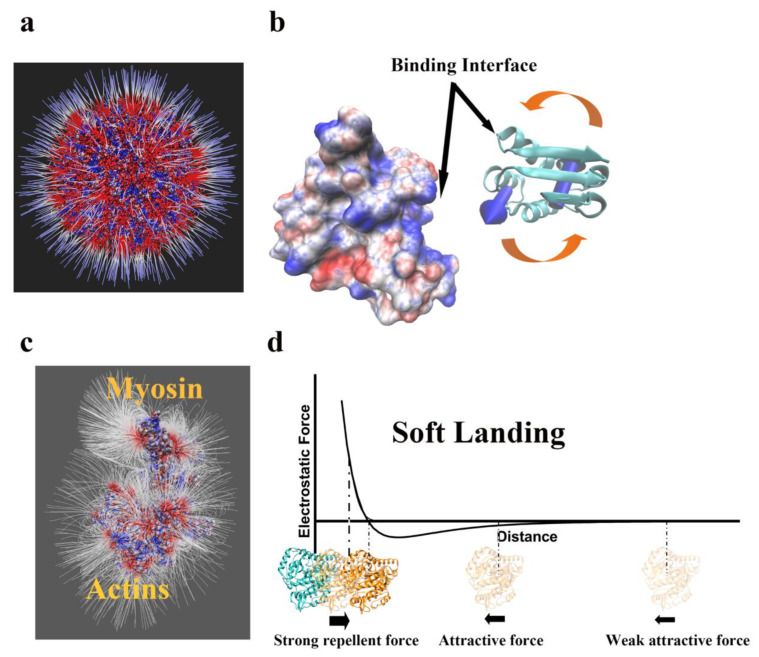
(**a**) Electrostatic features surrounding a viral capsid (PDB ID: 5J36 [59]). The electrostatic potential and electric field lines on the surface of the viral capsid illustrate the electrostatic interactions among the capsomer proteins. (**b**) The torque in barnase–barstar binding, where the barstar was separated by 20 Å and rotated by 90° to show the torque on the barstar. (**c**) The electric field lines on the myosin–actin complex, where the myosin and actin were separated by 15 Å to show the electric field lines. (**d**) The relationship between the electrostatic force and distance in protein–protein interactions. The blue one is alpha-tubulin while the orange one is beta-tubulin.

**Figure 3 ijms-23-10347-f003:**
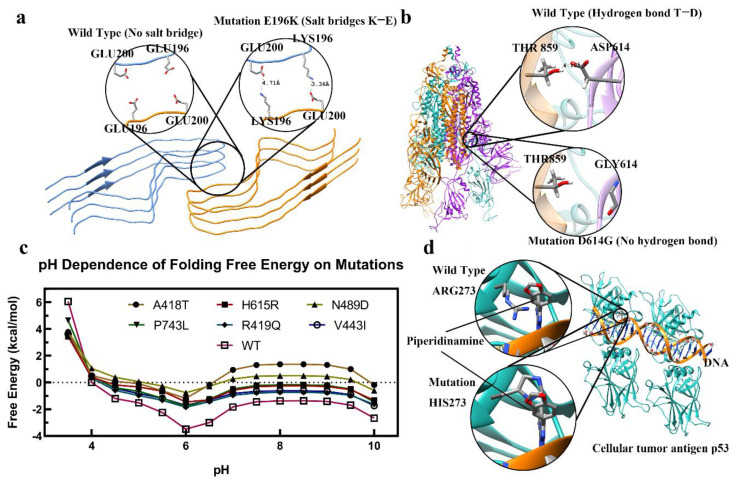
(**a**) Mutation of E196K formed a new salt bridge between GLU and LYS in the prion protein (PDB: 7DWV [99]). (**b**) Mutation of D196G (SARS-CoV-2 spike protein) caused the loss of the hydrogen bond between THR and ASP (PDB: 7KMS [100]). (**c**) The pH-dependence of folding free energy for the melanosome of oculocutaneous albinism-2 [8]. (**d**) The image shows p53 protein–DNA and 1-[2-(1,3-benzodioxol-5-yl) pyrazolo[1,5-a] pyrazin-4-yl]-3-piperidinamine. All mutations in a, b, and d are achieved using Chimera [101].

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
