# Peer review of "Electrostatics in Computational Biophysics and Its Implications for Disease Effects"

_ijms, 2022, doi:10.3390/ijms231810347_

Round 1

Reviewer 1 Report

I recommend that it can be accepted for publication after the following issues are addressed:

1. In this manuscript, the role of electrostatic interactions in biomolecule is reviewed. Considering the fact that similar words such as Coulombic interactions, electrostatic effects, electronic polarization, hydrogen bonds etc in the main-text, to avoid the ambiguity,  the concept of electrostatic interactions should be provided clearly in the first paragraph. 

2. What's the difference among electrostatic interactions, electrostatic force and electrostatic effects? In my opinion, short-range electrostatic effects is the mix effect arising from electrostatic interaction, orbital relaxation and Pauli-repulsion. Is it right? 

Author Response

Dear reviewer,

Thank you for the comments and suggestions for our manuscript. We have revised and improved the manuscript based on your comments. All the revisions made in the revised manuscript. The questions are addressed as shown below:

  1. In this manuscript, the role of electrostatic interactions in biomolecule is reviewed. Considering the fact that similar words such as Coulombic interactions, electrostatic effects, electronic polarization, hydrogen bonds etc in the main-text, to avoid the ambiguity, the concept of electrostatic interactions should be provided clearly in the first paragraph.

Response: As suggested by the reviewer appropriate text is inserted in the first paragraph of the manuscript (line 33-34).

  1. What's the difference among electrostatic interactions, electrostatic force and electrostatic effects? In my opinion, short-range electrostatic effects is the mix effect arising from electrostatic interaction, orbital relaxation and Pauli-repulsion. Is it right?

Response: “Electrostatic effect” term refers to any phenomena arising from electrostatic interactions, such as conformational change due to electrostatic interactions between charged atoms, proton uptake/release caused by electrostatic interactions, etc. The terms “electrostatic interactions” and “electrostatic force” are used interchangeable. The first one refers to interactions in general manner without specifying the direction and magnitude of the force, while the second is more specific and used when such an information (direction and magnitude of force) is available. 

Indeed, the short-range effects are results from the electrostatic interaction, orbital relaxation and Pauli-repulsion. This is added in the revision on line 153-155, first paragraph of short-range electrostatic effects.

Reviewer 2 Report

In the article, the authors review "resent contributions emphasizing on the role of electrostatics in the field of computational molecular biophysics and disease mechanism." The idea behind this review is good because electrostatics have key roles in protein structure and function, and the computational methods that have been developed help the field to understand the underlying mechanisms. The authors do a good job of highlighting some of the recent advances in this area.

However, the article requires extensive editing, for misspelled words, e.g. "resent" in the above referenced sentence (also "mechanism" should be plural), missing words, e.g., "states prior [to] any modeling", use of fragment sentences, e.g., "The interplay between favorable direct interactions and desolvation penalty.", unnecessary capitalization, e.g., "a salt bridge, The D263-K328 salt-bridge," and poorly structured sentences, e.g., "DFMD can successfully dock to barnase" that implies DFMD is the ligand. Also, some abbreviations were not defined, e.g., "MTBD". It would be helpful to provide brief descriptions of not-so-commonly-used terms/techniques, such as "Evald sum implementation" and "Drude Oscillator model". And what might be a benefit of a "soft landing" with respect to a binding interaction?

Author Response

Dear reviewer,

Thank you for the comments and suggestions for our manuscript. We have revised and improved the manuscript based on your comments. All the revisions made in the revised manuscript. The questions are addressed as shown below:

However, the article requires extensive editing, for misspelled words, e.g. "resent" in the above referenced sentence (also "mechanism" should be plural), missing words, e.g., "states prior [to] any modeling", use of fragment sentences, e.g., "The interplay between favorable direct interactions and desolvation penalty.", unnecessary capitalization, e.g., "a salt bridge, The D263-K328 salt-bridge," and poorly structured sentences, e.g., "DFMD can successfully dock to barnase" that implies DFMD is the ligand. Also, some abbreviations were not defined, e.g., "MTBD". It would be helpful to provide brief descriptions of not-so-commonly-used terms/techniques, such as "Evald sum implementation" and "Drude Oscillator model". And what might be a benefit of a "soft landing" with respect to a binding interaction?

Response:  Thank you for the suggestions. The manuscript has been refined to correct the plural, capitalization, and misspelling. The full names of all abbreviations are added in the corresponding paragraphs (line 90). “Evald sum implementation” and “Drude Oscillator model” have been explained, proper references were added (line 71 and line 170). Soft landing may help two binding partners to contact each other in sufficient time, so that they can adjust the configurations at the binding interfaces to avoid atom-atom clash and form hydrogen bonds and salt bridges. The description of the soft landing concept is added in the manuscript (line 129-132).